# Water Resources and Water Quality Assessment, Central Bamyan, Afghanistan

Hasan Ali Malistani [1,*], Hussain Ali Jawadi [1,2,*], Roy C. Sidle [3], Masuma Khawary [4,5] and Aziz Ali Khan [3]

1   Department of Geology, Faculty of Geosciences, Bamyan University, Bamyan 1601, Afghanistan
2   Afghanistan National Water and Environmental Research Center, Kabul 1006, Afghanistan
3   Mountain Societies Research Institute, University of Central Asia, 155Q Imatsho Street,
    Khorog 736000, GBAO, Tajikistan
4   Department of Biology, Faculty of Basic Sciences, Bamyan University, Bamyan 1601, Afghanistan
5   Microbial Pathogensis and Microbiom Laboratory, Department of Microbiology,
    Central University of Rajasthan, Ajmer 305817, Rajasthan, India
*   Correspondence: hmalistani@gmail.com (H.A.M.); hussainali.jawadi@gmail.com (H.A.J.)

**Abstract:** We surveyed and selectively sampled the major water sources in Bamyan city and the surrounding area to assess the water quality. Water quality measurements were taken in situ and more samples were collected for laboratory analysis from canals, rivers, springs, wells, and water supply systems. In urban areas, water supply systems provide 36% of the drinking water, but in rural areas, this source accounts for only 7% of drinking water supplies. Wells comprise 33% and 15% of urban and rural water supplies, respectively, while canals and rivers are modest water sources for Bamyan communities. Basic water quality parameters, such as pH, EC, and TDS, were variable with high values in some areas. Most of the samples fall in the range of potable water, but some had a high TDS and EC indicating that there is the potential of contamination. Values of pH were mostly were mostly in the range of drinking water (6.5–9.5). A Drinking Water Quality Index (DWQI) was calculated to better understand the water quality issues for the potable water supplies. Subsets of representative samples were analyzed for 17 selected chemical elements and other constituents. Barium (Ba) was detected in almost all of the water samples, while arsenic (As) was detected in about 9% of the analyzed samples, and this was mostly associated with thermal springs. Concentrations of Mn and Cu in some samples exceeded that of the water quality standards, while Zn concentrations were below tolerable limits in all of the samples. Most of the analyzed water samples were hard, and several samples showed evidence of microbial pollution in urban areas. Rivers originating from snow and glacier melting had excellent quality for drinking.

**Keywords:** water resources; WQI; hardness; pH; EC; TDS; Bamyan; Afghanistan

## 1. Introduction

Water is a fundamental and essential natural resource that mainly occurs in rivers, streams, lakes, glaciers, snow, rain, springs, and groundwater [1–3]. A water quality assessment includes the physical, chemical, biological, and environmental parameters of its contents, based on their concentrations or attributes being below defined limits [2,4–6]. The potential usage of various water supplies is greatly reduced when these are contaminated [7]. The accessibility and suitability of drinking water supplies are thus, critical livelihood issues, particularly in developing countries where groundwater is the major potable water supply and water treatment options are very limited [7–9].

Water resources have been assessed for drinking quality at basin, regional, and national scales in many countries [10–14]. However, such assessments in Afghanistan are sparse, and are undertaken only in larger cities like Kabul [10,15–18]. Due to a lack of high-quality data, these assessments have not always been beneficial [19,20]. Thus, in addition to robust water quality data, other environmental information is needed to assess the long-term drinking

water supplies in rural and urban areas. To ensure that proper assessments of available fresh water in developing countries are conducted, long-term records of precipitation, river discharge, reservoir levels, and status of groundwater resources are also needed [10,21].

Accurate information on national and regional water resources, both surface and groundwater, is needed to support sustainable economic and social development [22]. Water resource data are used in various ways within different government and local sectors for planning, development, and operational activities [23]. On the other hand, access to clean water plays a central role in achieving a country's development; nowadays, almost all actors are concerned that water supplies are becoming increasingly scarce [10,24]. In addition to water scarcity due to urban population growth, the degradation of water quality is a major concern [25].

Afghanistan has a population of more than 30 million with 74% of them living in rural areas. Access to clean drinking water is a critical issue because of the growing Afghan population, with only 57% of households having access to improved water sources [26–28]. Furthermore, only 31% of households have both improved sanitation facilities and only 21% have both improved water sources and sanitation facilities. These are some of the lowest rates, worldwide, thus reflecting the lack of safe water resources and the inadequate water infrastructure. More than 41,000 people, mainly children, die from diarrhea annually and 22% of deaths under 5 years old are attributed to diarrheal diseases. The infant mortality rate in the country is 10.3%, while the mortality rates of children that are 5 years old and under is 14.9% [28,29]. This situation has been exacerbated by the recent Taliban takeover following the US military withdrawal in 2021 [30].

Generally, Afghans acquire their drinking water from unprotected wells, rivers, and springs. Because of the occurrence of recent droughts and population growth, many cities and provinces of Afghanistan face reduced supplies and the contamination of drinking water [31]. These challenges will likely increase with projected future decreases in precipitation and higher temperatures [25,32].

The objective of our research is to quantify the water resources of the Bamyan District, as well as to assess some of rudimentary physical–chemical properties of these primary water sources. We then assessed the water suitability by developing a water quality index for Bamyan City and its suburbs.

## 2. Materials and Methods

### 2.1. Description of the Study Area

2.1.1. General Characteristics

Bamyan Province, which is located in the Central Highlands of Afghanistan, is one of the poorest provinces in Afghanistan, with most people living in rural communities [33]. According to a 2010–2011 report by Central Statistics Organization of Afghanistan, Bamyan Province had one of the highest rates of child mortality: 11.5–12.7% of live births. One reason for this high rate is the lack of clean water that is available for domestic use [34]. Bamyan Valley, which is located in the northern area of Kuhi-Baba Mountains (western-most part of the Hindu Kush and the upper-most part of Kunduz River watershed), mostly depends on surface water from melting of snow and ice. Higher elevations contribute the largest supplies of meltwater, with snowmelt typically occurring in late spring to mid-summer and glacial melt dominating from mid-to-late summer (Figure 1).

Historically, Bamyan City and the greater district had ample potable water supplies. However, with recent rapid urbanization and population growth, coupled with the construction of unplanned residencies and villages in hills and dry valleys, Bamyan City faces inadequate supplies of clean drinking water. There are different, but limited, available sources of drinking water for the population of Bamyan Central District (nearly 100,000 people) [33]. The district's demand for water supply is met by two main sources: surface water (rivers, canals, springs, and a glacier) and groundwater (wells) (Figure 2).

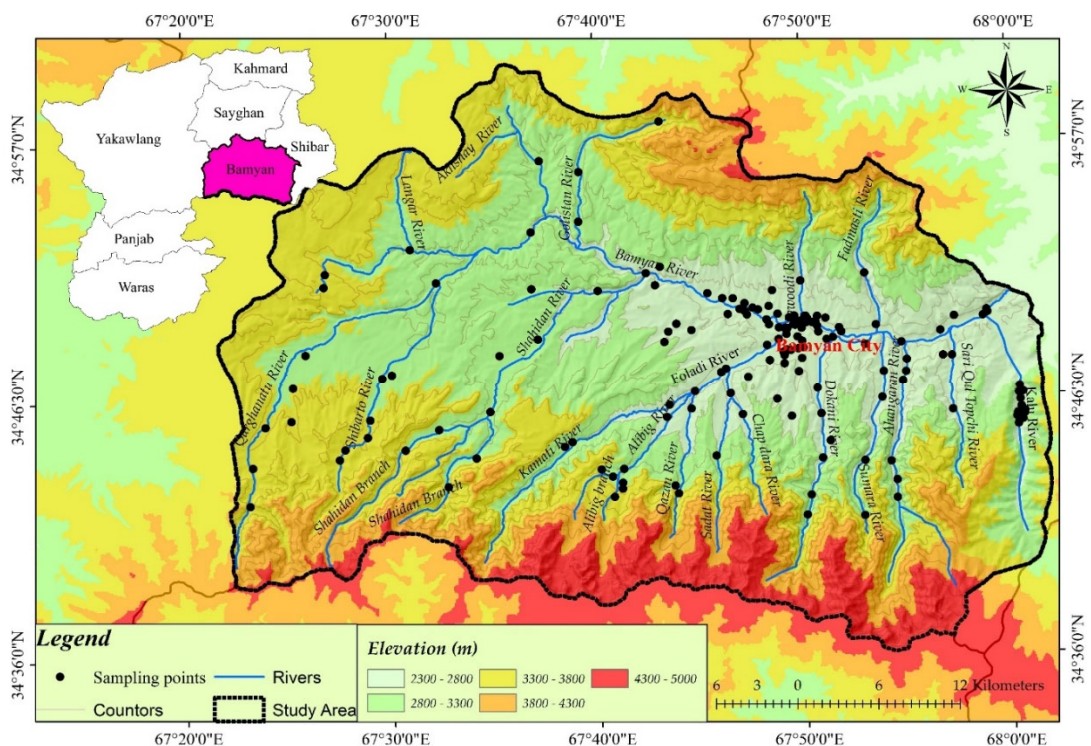

**Figure 1.** Study area in Bamyan Province, main surface water sources (rivers and springs), and locations of measurement points in Bamyan Valley.

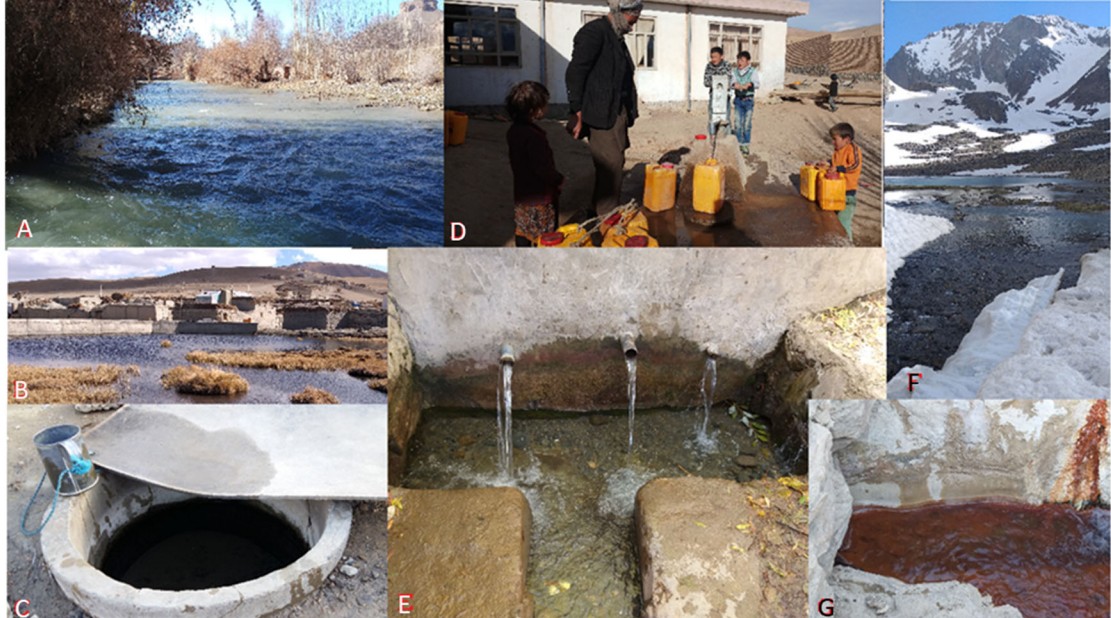

**Figure 2.** Examples of the main drinking water supplies investigated in Bamyan. (**A**) Bamyan River near Bamyan City (May 2020) as the main source of water for different uses in the district. (**B**) A water reservoir near Shibartu village, west of Bamyan. (**C**) An open well in Tulwara area in Bamyan (water table near the surface). (**D**) A community hand pump well in Sangchaspan. (**E**) A simple local water supply system in Sadat Valley, providing fresh water. (**F**) Shah-Foladi Lake at 4200 m above sea level; a major water source to Foladi and Bamyan Rivers via snow and ice melt (one of the purest sources of drinking for Bamyan City). (**G**) A thermal spring in Shahidanarea; one of the many hydrothermal brines flowing to Bamyan River.

Recently, groundwater increasingly provides drinking water for many urban areas, but technical challenges prevent the supply of this water to all district residents. These challenges include aquifer restrictions that are related to the conditions of basement rocks, the increasing costs of drilling, the poor-quality groundwater in some areas due to salt-laden and clayey substrates, and the contamination from deeper brines (Figure 2G). Moreover, groundwater is not a fail-safe resource as it may contain natural contaminates that are associated with the geology (e.g., arsenic) or bacteria that are introduced from septic system drainage in populated areas. While no studies have been conducted in the greater Bamyan area on potential contamination sources, a survey of the groundwater quality in the Kabul basin noted that contaminates are likely associated with poorly managed septic systems, leaking wastewater pipes, animal waste, and agricultural runoff [32].

### 2.1.2. Geological Setting

Bamyan Province has an arid and semiarid climate regime with dry/warm summers and very cold winters with temperatures of around $-20\,^{\circ}$C. The overall annual precipitation is about 165 mm [35]. The average elevation of Bamyan City is about 2500 m above sea level. Snow cover and glaciers of Kuhi-Baba Mountains are the main sources of groundwater recharge in the region (Figure 2F). From a geological perspective, Bamyan valley is very complex with a Precambrian basement, Paleozoic, Mesozoic, Tertiary and Quaternary formations. Metamorphic rocks such as schist, slate, quartzite, and gneiss outcrop on the southern hills, while magmatic intrusions are exposed and culminate at higher elevations of the Kuhi-Baba Ridge [36]. Neogene alluvial deposits and Quaternary moraines and alluvial fans and plains are major groundwater reservoirs in Bamyan Valley. The accumulation of fine materials, such as clay and carbonates, in the lowlands near Bamyan City creates a complex environment in relation to water quality [37].

The ridges in northwestern Bamyan primarily consist of sedimentary formations, mostly of carbonates and basic-to-intermediate volcanics of Carboniferous, Permian, and Cretaceous. The Triassic granitic batholith of the western Hindu Kush intermittently intrudes some of the older strata. This part of Bamyan Valley has limited water resources, and because of carbonates and clays, the water quality is poor [26,36]. The major Herat-Panjsher fault cuts through these rock formations and this is associated hydrothermal brines which appear in the eastern (Paymury) and western (Azhdar) areas of Bamyan (Figure 3).

### 2.2. Sampling

Water samples were collected from available and important sources of surface waters (water supply systems, rivers, streams, canals, reservoirs, and springs) and from groundwater (domestic wells) originating largely from snow and glacier melt throughout the entire Bamyan Central District and Bamyan City, including all the upper stream valleys. To limit the high turbidity effect of snow and glacier melt in the summer, the physical properties of the water were measured mostly in autumn (October-December 2018). In total, 217 samples (Figure 1 and Table 1) were collected from urban and rural areas to assess physical properties of the water. At the same time some common parameters (e.g., pH, EC, TDS, temperature, color, and turbidity) were measured in situ.

Also, 25 representative samples were collected from different key water sources (tube wells, dug wells, household supplies, and public water supplies), thermal and normal springs, and water supplies in urban areas for a more detailed laboratory analysis. Before sampling from open-dug wells, the water was pumped for 1–2 min to obtain representative samples. All samples were stored in 1000 mL sterile, dry glass bottles and kept in coolers with ice. These samples were then transported to the laboratory as soon as possible for a physiochemical analyses for an assessment of the potable water quality. These sample sites included eight wells, ten springs (hot and cold), and four water supply systems, which serviced water for domestic use. Because well water is the most important water supply for Bamyan residents, more samples were collected from the community wells and water

supply systems. In addition, critical sample points were established in the Bamyan River, Foladi River, and other streams, springs, and canals that recharge or discharge from the rivers. Potential areas of impaired water quality as well as the amount of various water sources were important criteria for the selection of these sampling points.

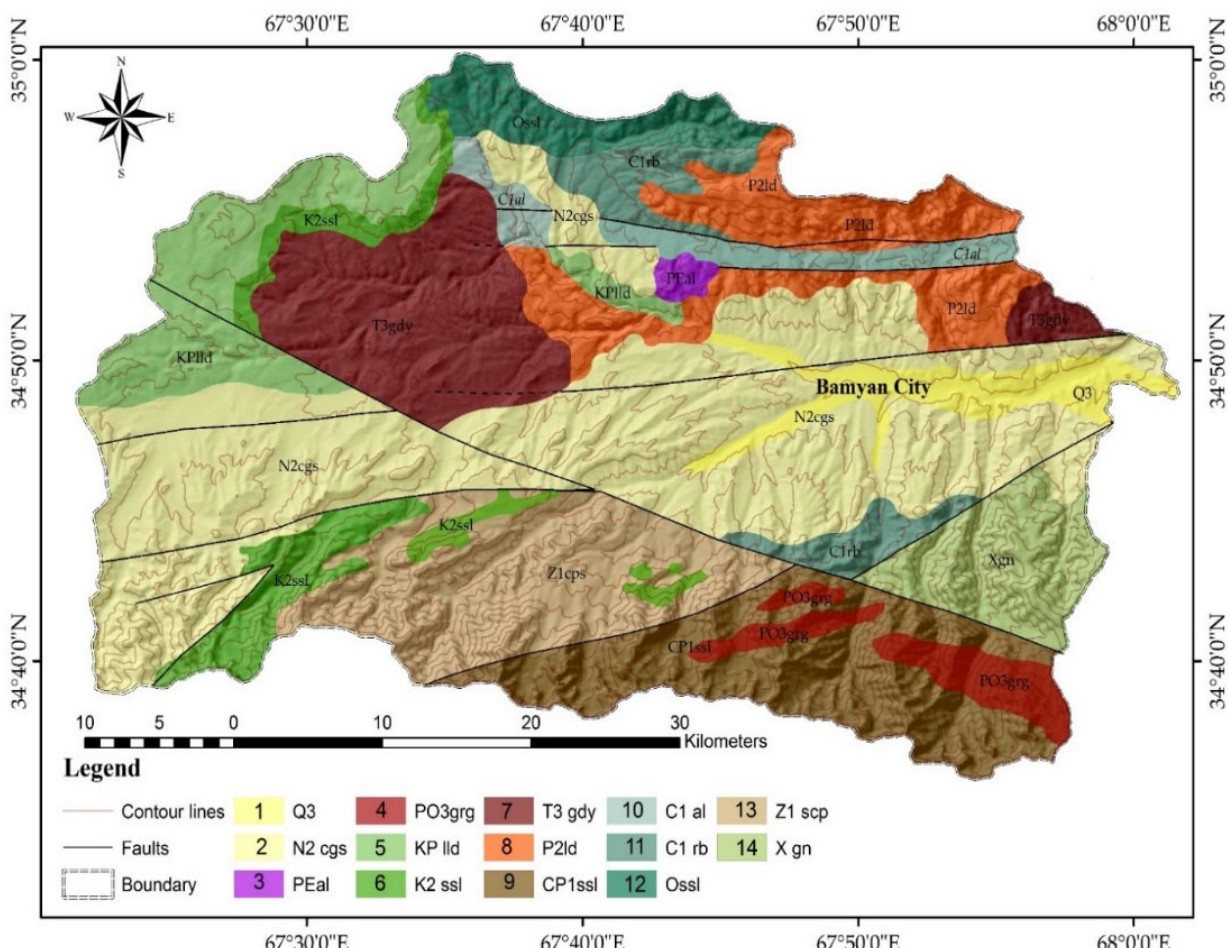

**Figure 3.** Geologic map of the study area. 1. Quaternary, alluvium, mostly gravel, sand, and silt. 2. Neogene, mostly conglomerate, sandstone and siltstone. 3. Eocene, andesite lava. 4. Paleocene-Oligocene, granite and granodiorite. 5. Cretaceous-Paleocene, limestone and dolomite. 6. Late Cretaceous, sandstone and siltstone. 7. Late Triassic, granite and granodiorite. 8. Late Permian, limestone and dolomite. 9. Carboniferous-Permian, sandstone and siltstone. 10. Early Carboniferous, andesitic volcanic. 11. Early Carboniferous, basaltic volcanic. 12. Ordovician, shale, siltstone, and schist. 13. Neoproterozoic, schist, slate, and phyllite. 14. Paleoproterozoic, gneiss (modified from [38]).

*2.3. In-Situ Measurements and Laboratory Analysis*

Selected physical properties, EC (HACH#8160), TDS (HACH-HQ), temperature (HACH#8156), color (HACH#8125), and turbidity (HACH#8237) were measured again in the laboratories of Bamyan University. Different EC, pH, and TDS meters were used in the field rather than in the laboratory. In most cases, the parameters were measured at least twice for each sample during the field collection period and at least four samples were taken to ensure that we achieved accuracy. The EC was measured as μS/cm and the TDS was measured as mg/L, with the EC and TDS values corrected to a standard temperature (25 °C). The EC and TDS were measured using pre-calibrated EC meters, and the TDS was estimated from the EC as follows: $TDS = 0.64 \times EC_{25}$. The pH meter was calibrated with a pH = 7.5 buffer solution.

**Table 1.** Details of the location and number of samples collected/measured, and available water resource types investigated in Bamyan District.

| Regions | Sampling Areas | N (Samples) | Number of Households | Water Resource Types Used (%) | | | | | |
|---|---|---|---|---|---|---|---|---|---|
| | | | | Water Supply | Well | Spring | Canal | River | Glacier |
| Urban | New Town (Isa Khan-Sar Asyab) | 16 | 2900 | 45 | 39 | 0 | 11 | 5 | 0 |
| | Old Town (Bazaar) | 22 | 1160 | 12 | 36 | 12 | 10 | 30 | 0 |
| | Sayed Abad | 5 | 2000 | 33 | 43 | 1 | 21 | 2 | 0 |
| | Molla Gholam-Azhdar | 9 | 1800 | 39 | 21 | 10 | 16 | 14 | 0 |
| | Rigshad-Sangchaspan | 11 | 1850 | 40 | 30 | 5 | 12 | 13 | 0 |
| | Zargaran-Jagrakhyl | 5 | 3450 | 44 | 27 | 0 | 21 | 8 | 0 |
| Rural | Syalayak | 4 | 480 | 0 | 9 | 18 | 28 | 35 | 10 |
| | Upper Foladi (Ali big) | 6 | 630 | 6 | 12 | 15 | 35 | 20 | 12 |
| | Lower Foladi Valley | 9 | 490 | 7 | 31 | 1 | 26 | 35 | 0 |
| | Qazan Valley | 5 | 330 | 0 | 8 | 25 | 27 | 25 | 15 |
| | Sadat Valley | 5 | 850 | 2 | 10 | 25 | 26 | 25 | 12 |
| | Khoshkak Valley | 7 | 280 | 12 | 32 | 19 | 20 | 17 | 0 |
| | DoKani Valley | 7 | 250 | 6 | 13 | 25 | 19 | 25 | 13 |
| | Sumara Valley | 4 | 150 | 0 | 14 | 14 | 31 | 29 | 12 |
| | Ahangaran Valley | 4 | 210 | 0 | 10 | 12 | 29 | 35 | 14 |
| | Tupchi | 4 | 640 | 16 | 17 | 18 | 19 | 30 | 0 |
| | Kalu Valley (Lower) | 7 | 530 | 8 | 8 | 17 | 29 | 29 | 8 |
| | Aqrabat-Gulistan | 5 | 250 | 10 | 10 | 20 | 20 | 30 | 10 |
| | Akhshai-Achaqul | 5 | 470 | 19 | 9 | 30 | 30 | 11 | 0 |
| | Qarghanatu | 6 | 230 | 6 | 31 | 6 | 31 | 19 | 6 |
| | Shibartu | 5 | 210 | 9 | 24 | 29 | 35 | 2 | 1 |
| | Shahidan | 10 | 390 | 2 | 14 | 24 | 29 | 24 | 8 |
| | Surkhdar-Khoaja Ali | 14 | 270 | 22 | 20 | 13 | 11 | 33 | 0 |
| | Fatmasti | 4 | 250 | 14 | 14 | 12 | 31 | 29 | 0 |
| | Bamyan River | 16 | | | | | | | |
| | Azhdar Springs | 7 | | | | | | | |
| | Paymury Springs | 15 | | | | | | | |
| | SUM | 217 | 20,070 | 353 | 483 | 351 | 567 | 525 | 121 |
| | % | | | | 15 | 20 | 14 | 24 | 22 | 5 |

Detailed chemical analyses were conducted on 25 samples that were collected from potentially problematic sites; these analyses were conducted at GreenTech, a commercial laboratory in Kabul. For the physicochemical analysis of the water, each sample was divided into three subsamples to determine the chemical elements and species, a biological analysis, and the measurement of As. Concentrations of Ba, Cu, total Fe, Mn, $SO_4^{2-}$, $Cl^-$, $NO_3^-$ and the water's hardness were analyzed by ion chromatography (MP-AES), while As was measured by a strep test (EZ Arsenic Test Kit) with very high accuracy. Some samples were collected from the urban area to assess the level of biological contamination, especially the total coliform levels; this was performed on the third subsample.

### 2.4. Development of a Water Quality Index (WQI)

Various approaches are available to assess the water quality, such as Prati's Index of Pollution, Bhargava's Index, Oregon WQI, NSF WQI, CCME WQI, Dinius' second Index, and the Weighted Arithmetic WQI [39–42]. The Weighted Arithmetic WQI is the approach that is most frequently used to classify water resources according to their appropriateness for drinking [39]. Some water quality parameters that are analyzed in situ or in the lab are used in the computation of the Weighted Arithmetic WQI. Each parameter is assigned a weight that indicates how important it is to the changes in water quality. The quality parameters are transformed into dimensionless values using a sub-index. The final weighted arithmetic mean of each sub-index is used to generate the overall WQI, which can range from 0 to more than 100 (Table 2). A higher WQI number denotes that there is a poorer water quality. To denote the differences, we used various colors to illustrate categories of water quality at the measurement sites (Table 2).

**Table 2.** Classification of water quality based on WQI values [43].

| WQI | Categories | Rank | Explanation and Potential Uses |
|---|---|---|---|
| 0–25 | Excellent | 1 | Drinking, Irrigation and Industrial |
| 26–50 | Good | 2 | Domestic, Irrigation and Industrial |
| 51–75 | Poor | 3 | Irrigation and Industrial |
| 76–100 | Very Poor | 4 | Irrigation |
| >100 | Unsuitable for drinking | 5 | Restricted use for Irrigation |

According to earlier studies [43–46], we calculated the WQI and classified the water resources using the following stages:

First, a weight ($w_i$) was given for each selected quality parameter based on its impacts on drinking water quality; parameters with greater negative impacts on water quality were assigned higher values, and those with lesser impacts were given lower values (Table 3). Because total coliform was not present in most of the monitored water samples, it was given a relatively lower weight. Sulfates alone do not impart serious health impacts, but when sulfate interacts with iron in water, toxicity can occur. Based on our field knowledge, we believe that the sulfur source for sulfate emanates either from barite ($BaSO_4$) or pyrite ($FeS_2$) given that the region is rich in iron and barium due to the nearby largest iron deposit (Hajigak iron deposit) in Bamyan. Based on this knowledge, sulfate was given a relatively high weight.

Second, the relative weight of each parameter was calculated by the Equation (1)

$$W_i = \frac{w_i}{\sum_{i=1}^{n} w_i} \tag{1}$$

where $W_i$ is the relative weight (weighted factor), $w_i$ is the given weight for each quality parameter, and n shows the number of selected parameters.

Third, using Equation (2), the sub-index of the water quality rating ($q_i$) is calculated as:

$$q_i = \frac{C_i}{S_i} 100 \tag{2}$$

where $q_i$ is the quality rating, $C_i$ as the concentration of each parameter in every water sample, and Si is the value of each quality parameter given by WHO.

Fourth, the sub-index of every parameter was calculated as:

$$SI_i = \sum W_i * q_i \tag{3}$$

where $SI_i$ is the sub-index of the $i$th parameter, $q_i$ is quality rating of the $i$th parameter, and $W_i$ is the relative weight of the parameter.



Finally, the WQI was calculated as follows:

$$\text{WQI} = \sum SI_i \tag{4}$$

**Table 3.** Drinking water quality input parameters, their WHO standard values, and weighting factors.

| Parameters | Unit | WHO Standard | Weight ($w_i$) | Weighted Factor ($W_i$) |
|---|---|---|---|---|
| pH | - | 6.5–9.5 | 4 | 0.0666 |
| Turbidity | NTU | 5 | 3 | 0.0500 |
| TDS | mg/L | 1000 | 4 | 0.0666 |
| Arsenic | μg/L | 10 | 5 | 0.0833 |
| Barium | mg/L | 0.7 | 4 | 0.0666 |
| Chloride | mg/L | 250 | 2 | 0.0333 |
| Cyanide | mg/L | 0.07 | 3 | 0.0500 |
| Copper | mg/L | 2 | 4 | 0.0666 |
| Hardness | mg/L | 500 | 2 | 0.0333 |
| Iron | mg/L | 0.3 | 4 | 0.0666 |
| Manganese | mg/L | 0.1 | 4 | 0.0666 |
| Nitrate | mg/L | 50 | 4 | 0.0666 |
| Nitrite | mg/L | 3 | 3 | 0.0500 |
| Sulfate | mg/L | 400 | 4 | 0.0666 |
| Zinc | mg/L | 3 | 4 | 0.0666 |
| Total Coliform | cfu/100 mL | 0 | 3 | 0.0500 |
| Fecal coliform | cfu/100 mL | 0 | 3 | 0.0500 |
| $\Sigma W_i$ | | | 60 | |

## 3. Results and Discussion

### 3.1. General Status of Water Resources in the Bamyan Area

The overall attributes of the study area, including the sub-area delineation, number of samples collected, number of households in each sub-area, and typical water resources that are used in each sub-area are shown in Table 1. To simplify, the study areas were sub-divided into urban and rural sectors. In each area, the water samples from different sites (sub-areas) were collected; a total of 217 samples were taken during this study. Surface and groundwater resources such as rivers, wells, springs, snow, and glaciers are utilized to supply water in both urban and rural areas. Water supply systems, which commonly use a combination of surface and groundwater, are the main sources of drinking water in urban areas, providing about 36% of the domestic urban water, while hand dug wells supply more than 30% of the domestic water supplies in urban areas. Rural water supplies varied across Central Bamyan, with rivers and canals providing the most amount of water. Additionally, rural residents receive drinking water through hand dug wells and springs. Due to relatively high quality of fresh river water that originates from snow and ice melt, these rivers (i.e., Alibig, Sadat, Qazan and Ahangaran rivers) are considered as the best source of water for domestic supplies.

In our detailed assessment of water quality, 17 physico-chemical parameters were considered (Table 4). Assessing the pH of the water samples revealed that the average pH was 7.21 (Figure 4). However, samples from most water sources exceeded the upper limits of turbidity, with a maximum turbidity of 57 NTU and an average value of 8.49 NTU. The TDS fluctuated significantly with a maximum value 5139 mg/L, which is very high compared to the WHO standard value (1000 mg/L), whereas the average value of the TDS (1050 mg/L) was also slightly higher than the WHO's acceptable value.

**Table 4.** Descriptive statistics of water samples in the study area. Units for all constituents are in mg/L except for As which is in µg/L.

| Parameter | Max | Min | Avg. |
|---|---|---|---|
| pH | 9.21 | 6.01 | 7.21 |
| Turbidity | 57 | 0.12 | 8.49 |
| TDS | 5139 | 61.57 | 1050.98 |
| Arsenic (µg/L) | 100 | 0 | 4.78 |
| Barium | 7 | 0 | 1.70 |
| Chloride | 1630 | 5.7 | 225.90 |
| Cyanide | 0.022 | 0.002 | 0.01 |
| Copper | 9.02 | 0.02 | 1.70 |
| Hardness | 2615 | 18.5 | 671.76 |
| Iron | 3.49 | 0.01 | 0.37 |
| Manganese | 2.88 | 0.007 | 0.42 |
| Nitrate | 1.11 | 0.004 | 0.10 |
| Nitrite | 17 | 0.5 | 3.47 |
| Sulfate | 1430 | 3 | 265.39 |
| Zinc | 1.18 | 0.08 | 0.31 |
| Total Coliform | 90 | 0 | 18.31 |
| Fecal coliform | 0 | 0 | 0 |

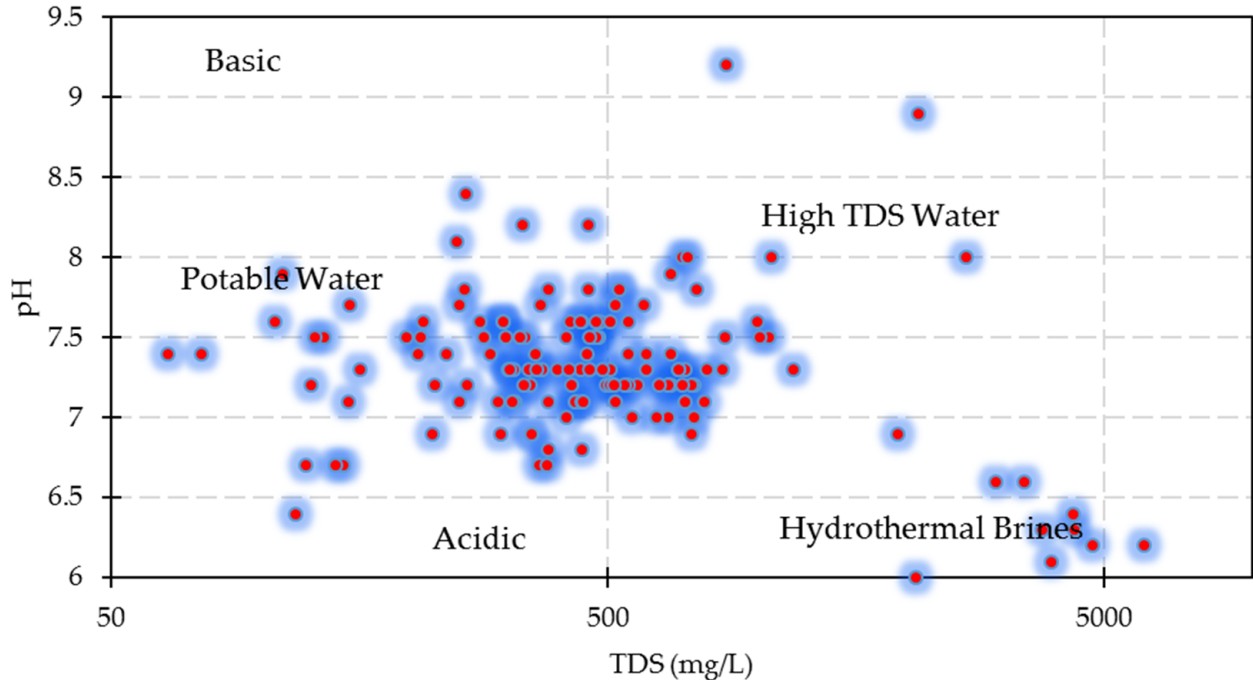

**Figure 4.** Relationship between TDS and pH in analyzed water samples.

Most of thermal springs within the study area were contaminated by trace elements such as arsenic, total iron, copper, manganese, zinc, and barium; the concentrations of these trace elements were typically higher than the acceptable concentrations for drinking water (average values: As = 4.78 µg/L, Ba = 1.70 mg/L, Cu = 1.70 mg/L, Mn = 0.42 mg/L, Fe = 0.37 mg/L, and Zn = 0.31 mg/L). However, high trace element concentrations did not appear in all of the water sources; for example, arsenic was only detected in about 9% of the

samples, while a small amount of the cyanide and zinc were recorded in the water samples from central Bamyan. In contrast, barium occurred in 96% of the analyzed water samples, and more than 30% of the water samples were contaminated by copper (Figure 5).

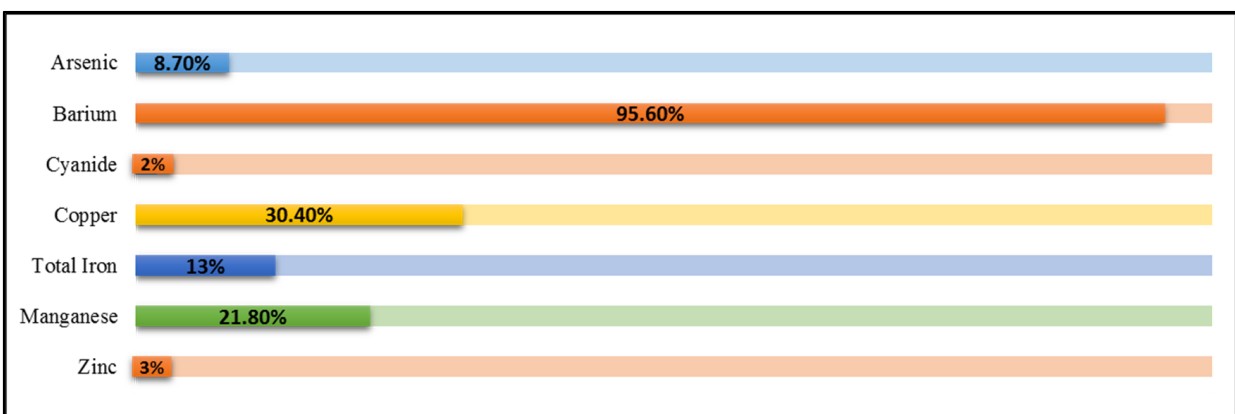

**Figure 5.** Percentage of samples containing measurable trace elements.

*3.2. Assessment of Water Quality Index (WQI)*

The WQI assessments for the water resources in central Bamyan are given in Table 5. The highest average WQI score (i.e., worst water quality) was observed in the waters of the Paymury thermal/hot springs (717.4). This elevated level indicates these waters are completely unsuitable for drinking. Normally, these waters would not be considered for drinking water, but in many cases, they feed into streams and rivers, thus compromising these drinking water supply sources.

The best water quality sources (lowest average WQI scores) were observed in the rural Ahangaran and Qazan valleys where most of the water originates from snow and glacier melt in areas with granitic basement rocks. The two rivers emanating from these valleys have excellent drinking water quality (WQI < 3.5); additionally, there are four other excellent quality rivers emanating from the upper Sadat (Jawkar), Sumara, upper Dokani and Ali Big valleys (WQI < 15). Most of the samples that were taken from wells and water supply systems have only fair water quality; wells in Molah Gholam have poor drinking water quality because these are mainly sourced from Azhdar salty springs. The water samples from lower Kalu valley, Khoja Ali, Qarghanatu, and Shahidan Rivers were categorized as poor quality with WQI values of 71.23, 65.12, 62, 59.22, and 60.71, respectively. The poor quality of water in Bamyan River can be attributed to its contact with anthropogenic activities as it flows through the populated old city of Bamyan (Bazaar). However, the poor quality of the lower reaches of Kalu, Dokani, and Sadat Rivers are due to the thermal spring chemical contributions. The water quality of the water supply systems was categorized as excellent, except for that in Azhdar because the water supply uses a salty spring source. Only the samples from the thermal springs showed that they contained completely unsuitable water for drinking. Such thermal waters are not directly used for drinking, but they can affect the streams and rivers that receive water from this source and that people sometimes use (e.g., in Azhdar and lower Dokani valleys). However, due to the lack of water treatment plants in the area, waters that were categorized as fair quality are also contaminated and likely unsuitable for drinking.

A closer examination of the results indicates that the current water quality in urban areas is somewhat sufficient and most of the water sources here have excellent to good quality. Exceptions included the wells in Mollah Gholam, the water supply systems in Azhdar, and the Bamyan River (Table 5). Like urban areas, the water quality from the main sources in rural regions (springs and rivers) is typically good to excellent. However, the lower reaches of the Kalu, Qarghanatu, Khoja Ali and Shahidan Rivers have poor water quality. Most of the wells and springs, except for the thermal springs (Syalayak, Dokani, Dahane-Ahangaran, and Shahidan), have mostly excellent quality water. This indicates

that the mixing of brines from the hot springs (e.g., Dokani, Syalayak, Dahane-Ahangran, and Paymury) affects the water quality in lower parts of the Bamyan Valley.

**Table 5.** General status of water samples and classification based on calculated WQI. Color coding reflects various WQI categories as noted in Table 2.

| Regions | Sampling Sites | No of Samples | WQI | | | |
|---------|---------------|---------------|-------------|-------|--------|-------|
| | | | Water Supply | Well | Spring | River |
| Urban | Old Town (Bazaar) | 22 | 8.38 | 13.83 | 5.76 | - |
| | New Town | 11 | 21.74 | 10.05 | - | - |
| | Bamyan River | 17 | - | - | - | 65.12 |
| | Molla-Gholam | 5 | 20.63 | 64.76 | - | - |
| | Azhdar | 4 | 71.43 | - | - | - |
| | Sar Asyab | 5 | - | 19.86 | - | - |
| | Sayed Abad | 5 | 21.87 | 10.62 | - | - |
| | Rigshad | 4 | 19.89 | 28.99 | - | - |
| | Sangchaspan | 7 | 21.09 | 30.65 | 19.87 | - |
| | Zargaran | 5 | 10.4 | 13.22 | - | - |
| Rural | AliBig | 6 | - | - | - | 18.91 |
| | Qazan Valley | 5 | - | - | - | 3.06 |
| | Sadat Valley | 5 | - | - | 12.52 | 4.56 |
| | Foladi | 9 | - | 4.18 | - | 32.67 |
| | Khoshkak Valley | 7 | - | 15.71 | - | - |
| | DoKani Valley | 7 | - | - | 11.72 | 14.9 |
| | Ahangaran | 4 | - | - | - | 3.12 |
| | Sumara Valley | 4 | - | - | - | 10.88 |
| | Tupchi | 4 | - | - | 19.03 | - |
| | Kalu River (Paymuri) | 7 | - | - | - | 71.23 |
| | Akhshay-Achaqul | 5 | - | - | 27.51 | - |
| | Aqrabat-Gulistan | 5 | - | - | 28.60 | - |
| | Fatmasti | 4 | - | 24.01 | 22.86 | - |
| | Khoja Ali | 10 | - | - | 17.03 | 62.00 |
| | Qarghanatu | 6 | - | - | - | 59.22 |
| | Shahidan | 10 | - | - | 11.46 | 60.71 |
| | Shibartu | 4 | - | - | 21.69 | - |
| | Surkhdar | 4 | 18.98 | 15.91 | - | - |
| Brines | Azhdar Salty Springs | 7 | - | - | 408.1 | - |
| | Paymuri Hot springs | 15 | - | - | 717.4 | - |
| | Dokani Hot Springs | 1 | - | - | 311.8 | - |
| | Shahidan Hot Springs | 1 | - | - | 371.1 | - |
| | Syalayak | 1 | - | - | 98.32 | - |
| | Dahane-Ahangaran Salty Spring | 1 | - | - | 567.6 | - |

Note: -, not applicable.

### 3.3. Correlation Matrix

Upon examining the correlation matrix for all of the variables (Table 6), some variables have significant positive correlations, such as turbidity versus Cu ($r = 0.89$); TDS versus Ba, $Cl^-$, hardness, Mn, $SO_4^{2-}$, and Zn ($r = 0.97$, 0.97, 0.99, 0.93, 1.00, and 0.95, respectively); As versus cyanide ($r = 0.81$), iron ($r = 0.81$) and $NO_3^-$ ($r = 0.81$); Ba versus $Cl^-$, hardness, $SO_4^{2-}$ and Zn ($r = 0.98$, 0.95, 0.98, and 0.93, respectively); $Cl^-$ versus hardness ($r = 0.93$), $SO_4^{2-}$ ($r = 0.98$) and Zn ($r = 0.90$); cyanide versus Fe ($r = 0.92$); and Zn versus $SO_4^{2-}$ ($r = 0.93$). A positive correlation indicates a similar origin for the variances, which may occur naturally or may be affected by anthropogenic activities [44].

**Table 6.** Correlation matrix for all of the variables.

| | pH | Turbidity | TDS | Arsenic | Barium | Chloride | Cyanide | Copper | Hardness | Total Iron | Manganese | Nitrate | Nitrite | Sulfate | Zinc | Total Coliform |
|---|---|---|---|---|---|---|---|---|---|---|---|---|---|---|---|---|
| pH | 1 | | | | | | | | | | | | | | | |
| Turbidity | −0.75 | 1 | | | | | | | | | | | | | | |
| TDS | −0.83 | 0.61 | 1 | | | | | | | | | | | | | |
| Arsenic | −0.43 | 0.47 | 0.54 | 1 | | | | | | | | | | | | |
| Barium | −0.78 | 0.59 | 0.97 | 0.56 | 1 | | | | | | | | | | | |
| Chloride | −0.76 | 0.60 | 0.97 | 0.55 | 0.98 | 1 | | | | | | | | | | |
| Cyanide | −0.76 | 0.75 | 0.83 | 0.81 | 0.83 | 0.85 | 1 | | | | | | | | | |
| Copper | −0.89 | 0.89 | 0.76 | 0.40 | 0.68 | 0.68 | 0.74 | 1 | | | | | | | | |
| Hardness | −0.85 | 0.61 | 0.99 | 0.56 | 0.95 | 0.93 | 0.82 | 0.79 | 1 | | | | | | | |
| Total Iron | −0.71 | 0.77 | 0.79 | 0.81 | 0.82 | 0.83 | 0.92 | 0.71 | 0.76 | 1 | | | | | | |
| Manganese | −0.82 | 0.59 | 0.93 | 0.62 | 0.89 | 0.84 | 0.80 | 0.76 | 0.96 | 0.75 | 1 | | | | | |
| Nitrate | −0.56 | 0.53 | 0.80 | 0.80 | 0.86 | 0.88 | 0.87 | 0.46 | 0.75 | 0.90 | 0.72 | 1 | | | | |
| Nitrite | −0.42 | 0.51 | 0.66 | 0.81 | 0.73 | 0.73 | 0.78 | 0.38 | 0.63 | 0.84 | 0.62 | 0.92 | 1 | | | |
| Sulfate | −0.80 | 0.61 | 1.00 | 0.59 | 0.98 | 0.98 | 0.86 | 0.74 | 0.98 | 0.82 | 0.92 | 0.85 | 0.71 | 1 | | |
| Zinc | −0.81 | 0.63 | 0.95 | 0.39 | 0.93 | 0.90 | 0.73 | 0.77 | 0.95 | 0.70 | 0.91 | 0.69 | 0.59 | 0.93 | 1 | |
| Total Coliform | 0.28 | −0.35 | −0.32 | −0.23 | −0.18 | −0.30 | −0.32 | −0.40 | −0.29 | −0.23 | −0.34 | 0.02 | 0.03 | −0.30 | −0.19 | 1 |

## 4. Conclusions and Recommendations

Drinking water supplies in Bamyan City and surrounding areas come from three main sources: wells, natural running water (springs and rivers), and municipal water supply systems. Among these sources, water supplies that use a mix of groundwater and surface water have a mostly acceptable quality and show the best water quality among the various sources. Wells typically have the next best water quality, and these are typically ranked from good to excellent. More than half of the water samples from the springs in Bamyan were not potable due hydrothermal contamination. However, all of the cold springs have good to excellent water quality.

Results from the surveyed areas, in situ tests, and the analysis of the water samples from the collection points showed important facts that need more interpretation and analysis:

(a) Geologically, we divided Bamyan District into two main basins: the southern and northwestern parts. Generally, the southern basin that is recharged by snow and glacier melt from the Kuhi-Baba Mountains has good water quality compared that of the to the northwestern basin that has different geology and topography with less snow and glacier coverage. The southern basin has metamorphic and magmatic basement hard rocks with little or no soil cover, a clayey substrate, and carbonate sediments. In contrast, the northwestern basin consists mostly of carbonate rocks and some shale, marl, volcanic, and granitic rocks. However, the role of bedrock and sediments on water quality is very complex and this needs a more detailed geochemical analysis and assessment to be performed. In addition, all of the basins showed a clear decreasing trend of water quality from the upper to lower reaches.

(b) Water supply systems, defined as simple piped water or reservoirs that recharge from springs, rivers, or wells, which have little protection and sanitation treatment, constitute only 15% of the water supplies to Bamyan communities, and these are mostly in urban areas (Table 1). These systems are operated largely by private companies and local communities and are too expensive for ordinary people to use. Many systems have failed to provide enough water for customers after several years and others failed during summer or winter. Due to the growing population and water scarcity in some areas during summer and winter, the need for the reliable regulation of public drinking water is evident, especially in urban areas. Water supply systems in rural areas remain primitive, as most are connected to unprotected canals or rivers. Therefore, we suggest the establishment of a central water quality monitoring authority within the Bamyan Water and Energy Department to regulate and control the public water supply systems.

(c) Dug wells, which are the second most important source of drinking water, are widely used in Bamyan. The water table is generally high in most parts of Bamyan; therefore, some wells reach water that is only a few meters deep. We surveyed wells that were from 1 m up to 50 m deep in different parts of Bamyan. It appears that, with time, the wells have been drilled deeper as more wells have become more developed. The quality of the water in these wells is generally good to excellent. Many of the wells were open and some receive inputs of surface water, while others were contaminated by natural brines or human activities. High levels of EC and TDS in the samples from many of the unprotected wells indicate that there was contamination from the walls of wells. We surveyed two types of wells—private and community wells. The quality of the water in the private wells is generally better than it is in community wells due to the overuse of the latter. In rural areas, where surface water is available, few wells are operational, especially where water supply systems are functioning.

(d) Canals are widespread, especially in rural areas, and they are basically used for irrigation. Many communities use water from these canals for washing and sometimes, if other sources are lacking, for drinking. Most of the canals originate from rivers and streams, with only a few receiving water from springs and reservoirs. The canals typically have the worst water quality, which is reflected in its color, turbidity, and

high TDS and EC. Due to this obvious high level of contamination, we tested a limited number of samples from these sources.

(e) Rivers are very important sources of water in Bamyan and almost all of the valleys in the region have a river or stream. The WQI of the rivers was mostly fair with some samples being in the good to excellent category. In the upstream areas where the rivers originate from fresh water sources like snow or glaciers, the water has a good WQI with low EC and TDS. However, the WQI in those that are downstream and near agricultural fields and populated communities is as low as it is in the canals. In some upstream rivers, we observed low pH (slightly acidic) water, which may be related to host rocks or basin sediments or mixing with hydrothermal brines.

(f) Fresh water sources, including springs and snow/glacier melting, are minor sources that only some rural areas directly use. Springs that are connected to glaciers or the upstream reaches of rivers have a good WQI with low EC and TDS values, but there are thermal, salty, carbonate, and other types of springs that have high EC and TDS values and a low pH due to the dissolved solids and metal content in them. Some hydrothermal brines (Paymuri and Azhdar springs) introduce considerable amounts of toxic elements (e.g., As and Ba) into rivers that change their quality from good to poor.

(g) Finally, based on the WQI assessment and accessibility, we suggest that the upper reaches of the Ahangaran, Sadat, and Qazan Rivers are the best sources for a central public drinking water supply system for Bamyan City.

**Author Contributions:** Conceptualization, H.A.M. and H.A.J.; H.A.M., H.A.J. and R.C.S., formal analysis, H.A.J.; investigation, H.A.M., M.K. and A.A.K.; resources, A.A.K. and R.C.S.; data curation, H.A.J.; writing—original draft preparation, H.A.J., H.A.M. and R.C.S.; writing—review and editing, H.A.J., H.A.M. and R.C.S., supervision, H.A.M., R.C.S. and A.A.K.; project administration, R.C.S. and A.A.K.; funding acquisition, R.C.S. All authors have read and agreed to the published version of the manuscript.

**Funding:** This study was supported by the University of Central Asia's (UCA) Mountain Societies Research Institute (MSRI) as part of UCA's "Pathways to Innovation" project funded by IDRC Canada and AKF-Canada.

**Data Availability Statement:** Most of the water quality data are presented in the paper. Any requests for the raw data can be made to the corresponding Afghan authors (H.A.M. and H.A.J.).

**Acknowledgments:** H. Malistani and H. A. Jawadi are thankful to the UCA staff for supporting Afghan scientists during the study, especially Bohdan Krawchenko for securing funding and supporting the study. Most of the water quality laboratory works were conducted in the Soil Science Department laboratory, Faculty of Agriculture at Bamyan University and we are grateful for the assistance from Z. Hashemi, T. Poyesh, and Abdul-Qahar. Numerous students from the Geology Department of Bamyan University participated in fieldwork and sampling, including Mohammad Reza Danesh, Husain Mosavi, Mahlegha, Sabera Mohammadi, Frishta Tawana, Halima Ebrahimzada, Hedayatullah Hedayat, Mohammad Esa Rafat, Basir, Saifullah, Mohammad Hanif, Gholam-Abbas, and others. We appreciate their contribution and assistance.

**Conflicts of Interest:** The authors declare no conflict of interest.

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
