# Peer review of "Water Resources and Water Quality Assessment, Central Bamyan, Afghanistan"

_water, doi:10.3390/w14193060_

Round 1

Reviewer 1 Report

The paper discussed the water resources and assessment monitoring in Central Bamyan, Afghanistan. Indeed, this is an important study in this geographical area for population health. I believe the idea is good and the date is interesting for this kind of study. To improve this manuscript these following remarks and suggestions:

·         The document needs to be better rewritten and structured. The abstract is poorly written

·         Introduction is also poorly written

·         The authors used long sentences:

* “Because only about 2.8% of the global water is fresh and suitable for human consumption and nearly 30% of this migrates to groundwater, which is a primary global source of drinking water, the accessibility and suitability of drinking water supplies are critical livelihood issues, particularly in developing countries where water treatment options are limited”

* Thus, robust assessments of water quality; long term records of precipitation, river discharge, and reservoir levels; and status of groundwater resources are needed to ensure proper assessments of available fresh water in developing countries (…)

·         The authors have not respected Template water: figure 1 …

·         Most of the references are forgotten (references from 28 to 37)

·         The authors have used information without citation to references

* Recently, undocumented reports have linked increases in certain cancers with water contamination in Afghanistan.

* Water supply systems, defined as simple piped water or reservoirs that recharge from springs, rivers, or wells, which have little protection and sanitation treatment, constitute only 15% of the water supplies to Bamyan communities, mostly in urban areas (…)

·         The data and methods used by the authors are good, but poorly exploited and poorly presented in the manuscript

Author Response

see attached response

Reviewer 2 Report

Paper is interesting for readers and very well manner presented by author.

Before Publication author should remove the English errors and improved the English language 

Author please some comments in attachment. . 

Author Response

see attached response

Reviewer 3 Report

Major water sources were surveyed and sampled for water quality assessment in Bamyan city and its suburbs. Water samples were collected from important surface water sources (canals, rivers and springs) and from wells and water systems.

The aim of the article was to describe the water resources of Bamyan district, as well as to assess some basic physico-chemical properties of these main water sources and to establish a water quality index for the city of Bamyan and its suburbs.

The article nicely describes the situation with water sources, the collection of samples, their analysis and finally the evaluation of their quality in relation to drinking water. The processing of the article is at a good professional level. The manuscript presents a high number of samples taken, 17 physico-chemical parameters were evaluated.

I have a few questions and comments:

Table 1: Please correct the term population (families) in the header of the table, perhaps a better term would be - number of inhabitants

Page 5 How to understand it, population (family) – why not inhabitant - how many members does the family have? that is such a strange data

Page 6, line 168: Finally, 25 samples were collected from different water sources .... Not 25 samples, this number present water sources from where water was collected for analysis

Page 8 Table 3: the WHO limit for pH is 6.5 to 9.5, according to which was the weight wi determined for individual parameters? It does not make sense. For example, total coliform bacteria has a value of 3, in my opinion it should have a higher value, why sulfates have a value of 4, what are the health effects of sulfates on humans? The manganese parameter also has a very low weight value, it should have the same weight number as iron. The Table 3 shows that the WHO limit for coliform bacteria is 10 cfu/100 ml, is this correct, shouldn't it be 0 cfu/100 ml? Can you explain that?

Domestic wells and individual population supply have milder limit values ​​than drinking water in public supply systems? Is this reflected in the article and the resulting assessment?

Page 10,  line 280: The  highest average WQI score (i.e., worst water quality) was observed in waters of the Pay- mury thermal/hot springs (717.4). This elevated level indicates these waters are completely unsuitable for drinking.   ...such water should not even be considered for drinking purposes, should it?

Page 12 Chapter 3.3: indexes in chemical substances need to be corrected, it is not written SO4-2 but SO42-

Page 13 It is necessary to change the numbering of the table, instead of 7 should be number 6

Conclusion: Do you have suggestions for improving the drinking water supply situation?

Attention: numbering of the literature! how is it in the text then? Please correct it and check it.

Author Response

see attached response

Reviewer 4 Report

The manuscript handles with the Water Resources and Water Quality Assessment, Central Bamyan, Afghanistan.

In the legend of figure 2 remove de reference “containing toxic elements like arsenic in drinking water of Bamyan City”, leave it only for the text.

Section 2.3: please add the brand of devices used to measure EC, TDS, temperature, colour, and turbidity.

Insert the brand for the kit of strep tests for As measurement.

Insert the device used for ion chromatography.

If I understand, from 217 collected samples, only 25 were analyzed with different major and trace elements. What samples were fully analyzed?

Table 4, I assume that the authors have mg/L, but as the name suggests they are trace elements usually in the range of micrograms and not milligrams as presented. On other hand putting the thermal springs before table 4 is very confusing. Why thermal are important here? Is the study for water quality of drinking water or thermal water?

The limit of detection and quantification for the different elements analyzed are fundamental and need to be inserted.

Author Response

see attached response

Round 2

Reviewer 1 Report

I am happy that authors improved the manuscript according to my comment. Still some small remarks here before the submission:

Please check again the template of water Journal. There are still some small mistakes. 

Please refer to the reference 30 in the manuscript.

what does it mean "15%15%" in the line 68?

Reviewer 4 Report

.